# Cooperation between Public Primary Health Care and Occupational Health Care Professionals in Work Ability-Related Health Issues

**DOI:** 10.3390/ijerph191911916

**Published:** 2022-09-21

**Authors:** Lauri Vähätalo, Anna Siukola, Salla Atkins, Tiia Reho, Markku Sumanen, Mervi Viljamaa, Riitta Sauni

**Affiliations:** 1Faculty of Medicine and Health Technology, Tampere University, 33014 Tampere, Finland; 2Health Sciences, Faculty of Social Sciences, Tampere University, 33014 Tampere, Finland; 3Department of Global Public Health, Social Medicine Infectious Disease and Migration (SIM), Karolinska Institutet, 171 77 Stockholm, Sweden; 4Pihlajalinna Työterveys, 33100 Tampere, Finland

**Keywords:** health care services, health care professionals, primary health care, occupational health services, cooperation, work disability, qualitative research

## Abstract

Work disability creates significant expenses for nations and causes human suffering by limiting patients’ lives. International studies show that to enhance recognition of and support for work disability, cooperation, mutual trust, and information exchange between public primary health care and occupational health care must be strengthened. However, little is known of how health care professionals experience this cooperation. The aim of this study was to understand how professionals experience the cooperation between public primary health care and occupational health services regarding patients’ work ability. Semi-structured interviews were conducted with 29 health care professionals working in five small cities (<10,000 inhabitants) in Finland. Interviews were audio and video recorded, transcribed verbatim, and analyzed through inductive thematic analysis. Three key themes were identified from the interviews: attitudes toward the other health care sector, the exchange of information, and resources for cooperation. Professionals seem to have poor knowledge about the services available and how care is given in the other sector, appearing to lead to weak mutual trust. The public primary health care professionals especially emphasized the benefits of cooperation, but several issues were mentioned as barriers to cooperation. These results can be used when planning effective patient paths and service provisioning models.

## 1. Introduction

Work disability occurs when a worker is unable to remain at or return to work because of a health problem. This causes not only large individual, social, and economic burdens, but also major concerns to workers, their families, employers, occupational health care providers, and society [1]. The costs related to work disability have been estimated to be rather high in European countries [2]. It has been estimated that solely in Finland, over 23 billion euros are annually lost due to diminishing work ability [3]. Therefore, it is important that the provisioning of work disability prevention services is continuously developed to react to emerging work disability challenges in order to help working-aged people to retain their work ability and to stay healthier later in life. Over 82% of working-aged people in Finland have access to occupational health care services (OHS) [4], which can identify and prevent work disability conditions early, but more effective cooperation between different health care sectors is needed [5]. Cooperation, and more precisely the interprofessional cooperation, is understood in this article to be established after the first information exchange aiming to enhance patients’ care is conducted between professionals working on different sectors [6]. 

Previous studies have shown that to be effective, health care services for the working-aged should include early recognition of work disability [7,8], enhanced communication between OHS and public primary health care regarding rehabilitation [9], and cooperation between health care sectors [10,11]. Cooperation is even more crucial when a person’s ability to work is compromised [12] or when the patient has been on sick leave for a long time [13]. Health care professionals have noted that cooperation allows better utilization of optimal care, since professionals have opportunities to consult each other and receive more detailed information about the patient’s profession, work environment, and overall situation [10]. Studies considering cooperation between health care sectors have highlighted the importance of understanding how professionals would prefer to cooperate and integrate care [5], how general practitioners (GPs) and occupational health physicians (OP) relate to one another [14], and the feasibility of the cooperation [15]. Previous studies have not considered how health care professionals experience cooperation in work disability prevention.

Three health care sectors provide primary care in Finland: public health care, funded by the state, including a service fee for patients; private health care, subsidized by the state, but mainly funded by patient service fees; and the OHS, funded by employers (75%) and employees (25%). Based on legislation [16], employers have to provide statutory OHS including preventative care, but it is voluntary to provide curative care for employees. The contents of the OSH contracts defining the curative services between service providers and workplaces vary greatly. Employees for whom employers have not organized curative care may use either public or private health care services to access care. A recent study showed that approximately 20% of occupational health care patients visited all health care sectors during a three-year follow-up [17]. This highlights the need for cooperation between sectors in order to support patients’ work ability [18]. 

Previous studies have shown that in contrast to occupational health physicians, GPs often lack the most recent information about the patients’ working and medical conditions [19], and relevant information considering fitness for work may be difficult to gather [20]. In addition, GPs seldom know the relationship between the health deficits and work [21] or have enough time to comprehensively evaluate the patients’ situation [22] to react to patients’ diminishing work ability accordingly. Therefore, the most relevant cooperation, communication, and patient guidance would ideally happen between GPs and occupational health physicians [19], and more broadly between public primary health care and OHS. However, little is known about how health care professionals regard this cooperation. Based on our best knowledge, this is the first qualitative study to examine cooperation on patient’s work ability issues between public primary health care and OHS. This study aims to understand how professionals experience cooperation between public primary health care and occupational health services regarding patients’ work ability.

## 2. Materials and Methods

### 2.1. Study Design and Study Setting

This is a qualitative study, which was conducted by using the hermeneutical phenomenological perspective [23] by utilizing inductive thematic analysis by Braun and Clarke (2006) to capture the very essence of participators’ experience [24]. This methodology requires the researchers to recognize and reject understandings that one has taken for granted and search the manifestations of beings disguised behind or within the spoken message [25]. However, the themes themselves resemble the presuppositions of the research since the language and understanding are formed on a prior structure [26]. To discover the true being of the participants’ lived experience, researchers must reflect continuously and profoundly their own thinking and dismantle the daily technical meanings by which they are prone to explain the living [25]. 

The study was carried out as part of a project which aimed to enhance the recognition of work ability deficits of working-aged people, develop patient guidance in work disability prevention services, and evaluate the effectiveness of these services. The five participating cities had the same service provider for OHS and public primary health care. 

### 2.2. Data Collection and Research Population

Participants were selected from public primary health care and OHS by using a purposeful sampling method [27]. The number of participants was determined in terms of sufficient information power [28]. The participants were recruited in public primary health care and OHS by the chief physicians and chief nurses, who were asked to invite those health care professionals who had worked continuously for a period longer than one year and whose patients were primarily working-aged adults. In Finland there are four primary professional groups on these sectors who treat working aged people: GPs and nurses in public primary care and occupational nurses and OPs in OHS. The data include 20 interviews with 29 participants. Originally 31 participants were invited, but one participant (a GP) declined the invitation, citing a hectic schedule, and one participant (an occupational health physician) did not respond to contacts by the researchers. 

All interviews were conducted via Microsoft (MS) Teams due to the COVID-19 pandemic, except one group interview which was held at the public primary health care center before meeting restrictions took place. Semi-structured interview guides (see Appendix A) were used, exploring the conceptions about patients’ work ability, patients’ service use, patient guidance, and the cooperation between public primary health care and OHS. Interview guides were piloted within the research group. No repeat interviews were done. All participants were sent an information briefing in advance, but the information was revised at the beginning of each interview. This was done to assure that the participants understood the interview context. All interviews were audio recorded and interviews done via MS Teams were also video recorded. All interviews were conducted in Finnish. Participants could attend without a video connection if it decreased the audio quality. Field notes were composed during the interviews to assure that all topics were covered, but the notes were not used for data gathering. The average length of the interviews was 33 min, the shortest was 21 min and the longest was 61 min. The data saturation was evaluated according to information power [28], and the research group regarded it to be sufficient. Interviews were transcribed verbatim and pseudonymized alongside the transcription. Transcriptions were not returned to the participants for commenting or correction. Transcription was done by a trusted third party, and sections marked as unclear or rich in content during the transcription were revised from the recordings. 

Of the 29 participants, 21 were women and eight were men. Most participants (n = 24) worked in public primary health care (12 GPs and 12 nurses) and five (one occupational health physician and four occupational nurses) worked in OHS. Participants may have had different specializations (e.g., nurse specialized in substance abuse, nurse specialist in diabetes treatment), but to ensure the anonymity of participants they are referred to as nurses, according to their initial education. Participants represented five different cities and seven health care organizations. The complete data comprise 14 individual interviews, four pair interviews, one interview with three participants, and one interview with four participants. Group interviews were preferred when interviewing nurses, since it was more appealing for the participating public health care centers and OHS to use one specific moment for the interviews than multiple separate occasions. The interviews were conducted between September and December 2020. Detailed information about the interviews and participants is presented in Table 1.

### 2.3. Analysis

The research team participated actively in refining the themes and codes and selecting citations; L.V. was most engaged with the coding and analysis. The inductive thematic analysis was performed following the method of Braun and Clarke [24] without following any theoretical framework [29]. Firstly, the data were read and reread, and initial codes created that formed the preliminary themes. This was followed by confirming the codes and themes, renaming them when necessary, formulating the thematic map (see Appendix A), and finally reporting them. Codes, themes, and the coding tree were formed inductively during the analysis, without relying on any preceding theory. An example of the analysis is displayed in the Table 2. The data and analysis were managed with Atlas.ti software (version 9.0.24.0 by Scientific Software Development GmbH, Berlin, Germany). The analysis and supporting quotations were translated into English by the research team when reporting the results.

### 2.4. Reflexivity

Researchers L.V. (PhD student) and A.S. (PhD) working at Tampere University had the main responsibility for the interviews. Both have previously worked on questions relating to work ability and health care services. L.V. is a social scientist and A.S. is a public health researcher. All interviews were conducted by L.V., but A.S. attended the majority of the interviews (14 out of 20). She assured that the interviews followed the semi-structured interview guides and were coherent.

## 3. Results

Three main themes were identified in the data: attitudes toward the other health care service sector, the exchange of information, and resources for cooperation (Table 3).

### 3.1. Attitudes toward the Other Health Care Service Sector

#### 3.1.1. Knowledge about the Other Health Care Service Sector

Most public primary health care professionals thought that the knowledge about OHS services and procedures was low in their sector. This was evaluated to negatively affect patient guidance and finding suitable work disability prevention services. Some professionals in OHS thought that the work ability assessment might be forgotten by accident in public primary health care: 


*Oh well, the thing that I think is the most central is that work ability would even be remembered. I think that it would be the most relevant point.—I don’t think that it is that difficult since the assessment of work ability is such a basic procedure for a physician, if they only think about it.*
(OHS, Occupational health physician, City 2)

The quotation displays the general thought among OHS professionals that public primary health care professionals do have the capability to assess patients’ work ability, but for some reason they tend to neglect or forget to do so. Thus, the problem of not assessing work ability was regarded to be associated with the way of working and processes in public primary health care, not to the incapability to carry out work ability assessment as such. 

The patients’ varying contents of OHS contracts were mostly unknown by the public primary health care professionals. This was regarded to cause difficulty for patient guidance and service provisioning. The problem was recognized also by OHS professionals: 


*When the patient calls the public primary health care center, the patient is instantly told to visit OHS, and if the issue requires medical care, the OHS contract may not cover it.*
(OHS, Occupational nurse, City 2)

The occupational nurse seems to assume that public primary health care personnel do not bother to find out whether the patient’s OHS contract covers the required treatment. This represents how knowledge and trust are combined. The OHS nurse does not seem to know the restrictions that public primary health care professionals face when they try to verify the contents of a patient’s OHS contract:


*And since occupational health care is such a complicated system and it is so different, depending on the current contract you may be unable to check anything, and there is no clear guidance for it.*
(Public primary health care, Physician, City 2)

Nearly all participating public primary health care professionals in every health care center regarded OHS to be a difficult system. This may affect how the role of OHS is seen as a care provider. This difficulty was contributed to further by the poor or completely lacking health information of patients who came to public primary health care from OHS.

Neither the public primary health care professionals nor the OHS professionals made remarks considering the daily patient work in the other health care sector. This fortifies the perception that professionals seem to be unaware of how care and services are provided to patients in the other health care sector.

There seem to be some similarities but also differences in what information is available for each sector. In OHS, there appeared to be a lack of knowledge about what the terms of clinical work and care given in public primary health care are, whereas the lack of knowledge about OHS in public primary health care was related to the contents of patients’ OHS contracts and to OHS as a system.

#### 3.1.2. Trust in the Other Stakeholder’s Services

The preceding subtheme showed how trust was based on knowledge, and without knowledge the formation of trust is hindered. Knowledge and trust also relate to cooperation, since without knowing suitable services, it is difficult to guide the patient to the most convenient care:


*Then, well, I don’t really know if public primary health care has anything to offer really at the moment, so yes, the process works well when the control is in our own hands and the process can be run through our own house, which is quite logical, since you know your own services, but right away when we go to the other side of the sector line (public primary health care), the picture gets blurry in every direction. It doesn’t really mean that the services there are bad, but I don’t really know them.*
(OHS, Occupational health physician, City 2)

Problems in cooperation and trust are not based merely on reserved attitudes about the other sector, but also on the feeling of losing control. This can be seen as a consequence of not being aware of what is being done to the patient on the “other side”. The very same problem was recognized in public primary health care, where patients sometimes just seemed to appear at the health care center, without any health information from OSH. GPs said that sometimes they could not do more than just hope that the patient brings the relevant information.

### 3.2. Exchange of Information

#### 3.2.1. Information Exchange Conventions

A requirement of effective information transfer concerns knowledge of how and why the other service sector works. In addition to the major lack of knowledge about the contents of a patient’s OHS contracts, GPs reported having problems with receiving information about their patients from OHS.


*I don’t necessarily see even the texts written by occupational health care, what has already been done, the care paths are not clear and the patients don’t know who to contact or address their problem to.*
(Public primary health care, Physician, City 2)

The patient being the main source of information was repeated in many interviews by the public primary health care professionals. It seems that GPs are highly dependent on the information that the patient provides, since the information is not systematically available in the Finnish Patient Data Repository (KANTA services), which is supposed to gather all patient data into one place. Some interviewees questioned the trustworthiness of patients as information agents and discussed patients’ ambitions to hide some parts of relevant health information. 

Contrary to the public primary health care professionals, the OHS professionals did not discuss problems related to information availability or transfer. The main information exchange issue reported by the OHS professionals was a lack of information about rehabilitation groups and other public rehabilitative services available for citizens.

In this study, public primary health care and OHS seem to be unequal in terms of information exchange. Deficits in information flow from OHS to public primary health care are described to affect the care given in public primary health care. In turn, the poor information flow from public primary health care to OHS seems to limit only the provisioning of additional services. Another related issue is the possibility to directly contact professionals of the other sector:


*I think we have a paper somewhere that has all the phone numbers… (searches the desk) I’m looking through the lists in my room to see whether there is a number to occupational health care anywhere... I don’t see it in the lists that are available to me here.*
(Physician, Public primary health care, City 3)

The lack of direct phone numbers to contact colleagues in OHS was seen as a concrete obstacle for cooperation in public primary health care. However, this was not true on every occasion, and there seemed to be differences between cities and between professions in public primary health care. For some unknown reason, public primary health care nurses appeared to more often have direct phone numbers or email addresses to OHS, while only one of the GPs reported having a direct personal phone number of a colleague working in OHS. The meaning of personal relations to contact colleagues working in the other sector remained unclear. Overall, OHS professionals seemed to have more opportunities to make direct contact: 


*I have a habit of asking for permission from the patient and we may call straight to the nurse specialized in drug abuse (located in public primary health care) for example, so I tell the nurse that the patient is with me in the same room and ask if it would be possible to get some support for the patient, or we may call even to the adult counseling center (located in public primary health care).*
(OHS, Occupational nurse, City 2)

Most OHS professionals mentioned more than one possible contact with public primary care. Only nurses in public primary health care discussed having the possibility to contact OHS, but neither they nor the GPs reported the possibility to contact multiple different professionals in OHS.

#### 3.2.2. Need for Changes in Cooperation

The public primary health care professionals described multi-professional cooperation to be beneficial in a situation where the patient’s health issues were related to work or the work ability was compromised. They also seemed to acknowledge the OHS professionals’ experience to handle issues related to work ability: 


*I think that they (OHS) have high expertise in these matters and sharing of that knowledge would be beneficial overall.—It would also enhance the service structure.*
(Public primary health care, Physician, City 3)

The public primary health care professionals appeared to be keen to learn more about OHS, but a lack of time hindered the opportunities to do so. Multi-professional cooperation was discussed several times as an alternative to time-consuming guidance, especially by the public primary health care professionals: 


*It’s just, if we think of work ability issues, they aren’t solved at the physician’s visit, rather it requires multiprofessional expertise and often cooperation between occupational health care and special health care.—It is more preferrable to schedule a multiprofessional meeting, where the focus is the patient, and this is more efficient for every participating party.*
(Public primary health care, Physician, City 1)

The benefits of a multi-professional cooperation meeting were regarded to outweigh the effort required to arrange it. Several GPs did propose multi-professional meetings as a preferred form of cooperation in work ability issues. However, the proposal was not in line with the GPs’ descriptions of their full timetables, which they reported to cause difficulties in making a single consultation phone call regarding the patient’s matter.

The OHS professionals did not discuss the need for multi-professional cooperation as often as the public primary health care professionals did. On the contrary, the OHS professionals seemed to have rather good consultation opportunities. They noted that occasionally a specialist visits their unit, and they can guide patients to see the specialist if the patient’s OHS contract covers it. 

Several OHS professionals brought up the necessity to clearly separate the care responsibilities between public primary health care and OHS: 


*Sometimes it is more essential that there remains some kind of border or limit between the sectors to remember that public primary care and occupational health care are separate operators, even though the same private service provider is providing both of the services.*
(OHS, Occupational health physician, City 2)

The notion that borders should be clear was not brought up by any of the public primary health care professionals; instead, some of them argued that bureaucratic lines hampered communication and cooperation: 


*And even though we are in the same building and even though we meet occupational physicians, for example, in the canteen where we have lunch, we don’t agree or discuss shared patients.*
(Public primary health care, Physician, City 3)

Some professionals from both sectors discussed the physical closeness of the other sector as a benefit since patients could walk from one health care service to another. However, they did not mention any other benefit regarding the closeness of the units. 

### 3.3. Resources for Cooperation

#### Feelings of Insufficiency

Money as a resource was mentioned in the dataset only twice, first in the context of cities’ health care budgets, which were thought to have decreased over time. The second time was when OHS nurses discussed how small private companies wish to limit their employees’ OHS use to save money. Instead of money, hurrying and a feeling of insufficient time to perform one’s job were the most apparent notions in this theme. These issues appeared in the interviews from both sectors. In public primary health care, the feelings of hurry and strict timetables were regarded as an obstacle for cooperation and communication with OHS:


*Since we have not received any time for the paperwork, I do it all within the one hour: I define the status, I investigate the background and make a care plan and maybe some sickness statements. All this within the one-hour appointment, since we don’t have time to do this, thus I don’t have any time to make any phone calls within that time. The time is limited.*
(Public primary health care, Physician, City 3)

An additional phone call, or any kind of connection to a colleague in OHS seemed impossible in the views of many public primary health care professionals. However, this was not in line with a willingness to cooperate. Those who declared that time was too limited to contact colleagues in OHS often addressed the value of cooperation or even multi-professional meetings. Professionals in OHS did not describe the feeling of hurrying as a problem with hectic timetables; in contrast, they felt it more as a feeling of insufficiency. The shortness of the health care visits (often 15–20 min) was especially reported as an obstacle for treating the patient accordingly.

The rapid change of health care professionals was mentioned to concern mainly public primary health care, and the topic was brought up without any specific question. GPs described this to be the major reason why shared procedures and guidelines are easily forgotten in public primary health care. This notion emphasizes the importance of introducing new staff, on the one hand, and it underlines the poor availability of shared guidelines, on the other. In addition to the poor utilization of shared procedures, the changing of the health care staff also creates a challenge for establishing cooperation between OHS and public primary health care: 


*And the physicians change in the public primary health care centers; you can’t keep track of those who are currently working in the health care center. Previously you could know exactly that there were those permanent physicians. The people change and are replaced so often that it also bothers the cooperation and advancement of the cooperation.*
(OHS, Occupational nurse, City 2)

## 4. Discussion

The study showed that the public primary health care and OHS professionals working in small Finnish cities perceived the cooperation between sectors as beneficial but difficult to establish. Three main themes were identified that describe the present cooperation circumstances: attitudes toward other health care services, the exchange of information, and resources for cooperation. It appeared that professionals in OHS have more opportunities to contact public primary health care than vice versa. In addition, professionals working in OHS discussed fewer difficulties with information flow between sectors than the public primary health care professionals. The insufficient time resources were also described to affect the care given in OHS less than in public primary health care. It was also found that poor knowledge about services, care procedures, and limited trust in the other health care sector hindered cooperation, even though public primary health care and OHS were provided by the same private health care provider in the participating cities. The physical proximity of the health care units—even when sharing the same building—did not foster cooperation between public primary health care and OHS. The situation where public primary health care and OHS are provided by the same private health care company is not common in Finland overall, but it exists especially in small cities (<10,000 habitants). 

Professionals from both sectors expressed a poor understanding about work in the other sector. In a Canadian study, the poor knowledge and trust of the other stakeholder has been recognized as an obstacle for cooperation in the context of workers returning to work [30]. In the Finnish context, it has been recognized that health care professionals lack knowledge about other service sectors (e.g., social care) [5,31]. In addition, deficits in GPs’ knowledge about OPs work have been highlighted in previous research in Germany [32] and in France [6]. This study suggests that professionals in public primary health care and OHS have deficits in knowing about the work performed in the other health care sector. In addition, the difficulties that the lack of availability of OHS contracts causes in public primary health has not been documented before. 

The analysis showed that knowledge, communication, and the building of trust between professionals are often intertwined [21,32]. The shared curriculums and joint education of GPs and occupational health physicians has managed to produce only minor and temporary changes in attitudes toward the other health care sector [33,34]. Thus, when aiming to fortify the cooperation between public primary health care and OHS, the barriers stemming from the service structure [35] could be avoided by increasing the initial knowledge and know-how about the other health care sector [13]. Therefore, future research and interventions should consider how local pre-existing connections and services could be used to support cooperation. In Finland this is even more relevant since the health care reform will change the health care provisioning model in the near future. 

Finland has a unique OHS, where curative care is integrated into the preventive work of OHS. Globally, there are various models of organizing OHS and the co-operation with public primary health care varies from country to country. The provision of OHS is specified by law in Italy and is integrated with primary health care. On the contrary, in Germany, OHS organizations operate under free market conditions and do not form part of a public health scheme or national health service [36]. However, the challenges in the cooperation in work disability prevention in the health care systems are common everywhere [6,32]. At European Union level a strategic goal has been agreed upon to enhance cooperation between OSH, employment and health experts, and competent authorities to take overall account of the opportunities, challenges, and needs related to the guidance, treatment, rehabilitation, and return to work of workers [37].

This study discussed the sectors’ different opportunities to make direct contact with the other health care providers in issues concerning work disability. Professionals in OHS and nurses in public primary health care described having used direct contact (e.g., phone and email) for consultation purposes more often when compared to GPs in public primary health care. There is pre-existing evidence from the United Kingdom suggesting that OHS professionals make more phone calls to public primary health care [9] than vice versa. This is confirmed in a German study that reported the most frequent mean of communication between OP and GP to be telephone [10]. In a Finnish study it was reported that especially GPs lack time to make calls to OHS [38]. Another study conducted in United Kingdom showed that OPs appeared to prefer sending a letter by post over telephone [39] However, the different opportunities of public primary care nurses and GPs to have contact to OHS has not been previously reported. Experiences about information transfer varied in this study, and some GPs even questioned the patients’ active role in information transfer between services. It has previously been reported in a Belgian study that occupational physicians prefer to deliver information through the patient to other health care sectors [13]. 

The finding that public primary health care suffers from unreliable information transfer from OHS is in line with a previous study from the United Kingdom [40]. This was the opposite of the perceptions of OHS professionals, who did not report problems related to information transfer. In a French study it was found that OPs seldomly encountered difficulty in information transfer, but they highly emphasized that professionals shared understanding about the patients’ situation across health care sectors [6]. Our finding that public primary health care faces difficulty in information exchange might be partially explained by Finnish data privacy legislation. It allows patients to choose whether the information in occupational health care records is visible to professionals in public primary health care [41].

The unexpected appearance of a patient with poor background medical information from OHS to public primary care may create skepticism about the care given in OHS, diminish trust between sectors, and raise questions about OHS procedures. This type of experience of isolation due to poor information exchange is a form of siloing, which has been related to the discontinuity of care [5], fragmentation of care [42], and mistrust between professionals [13]. As a consequence of the out-sourcing of health care services at the county level, more coordination and better information flow between health care service sectors has been emphasized in order to reduce siloing [5].

The experience of insufficient time differs between public primary health care and OHS. In public primary health care, a lack of time was reported to affect opportunities to conduct work optimally (e.g., make consultation calls and patient guidance), thus limiting the work of health care professionals [38]. In OHS, the insufficient time seems to appear more as a feeling of deficient service provisioning, such as short reception times. Insufficient time may also be the reason for poor cooperation between sectors, at least from the public primary health care perspective [6,38]. This raises the question of whether patients who have only statutory occupational health care receive as high-quality and consistent care with respect to their work ability, since the available resources to assess the work ability in public primary health care are scarce [21]. In addition to the poor time resources, public primary health care professionals tend to change working places often, and this creates notable difficulties in establishing and maintaining cooperation between sectors [38].

Due to the recent out-sourcing trend in health care services [43] and in regard to the up-coming health care reform in Finland [44], the results of this study are relevant both in describing the current status of cooperation, and also in supporting decision making when the reformed health care is planned. The willingness to cooperate in issues related to work ability could be utilized in concrete actions on a local level, since local health care providers have the best knowledge about the current needs and possibilities for cooperation between health care sectors. However, this would need strategic decisions to guide the cooperation and emphasize the importance of work disability prevention.

Future studies should investigate what legislative or political decisions should be made to enable cooperation between public primary health care and OHS. On a national level it would also be interesting and meaningful to know whether the willingness to cooperate differs between areas and different sized cities or units. A national survey and interviews that include different public primary health care and occupational health care units would be plausible when researching the differences in local health care settings. In addition, research regarding the cooperation between other health care sectors (e.g., between private and OHS or OHS and special health care) could reveal good practices to be utilized in intersectoral cooperation. Interviews and interventions would serve this cause the best, since interventions can reveal obstacles and practical solutions that are not visible outside the organizations. Overall, cooperation between health care sectors is a phenomenon that could be approached more from the intervention research perspective.

### Strengths and Limitations

The interviews provided deep and nuanced perceptions about cooperation on patients’ work ability issues between public primary health care and OHS. While there was an imbalance of OHS and public health care sector representatives, this reflects the unit sizes in the participating cities (<10,000 inhabitants). Thus, the results represent the health care professionals’ perceptions in this type of health care provisioning setting where the OHS and public primary health care are provided by the same health care company. Further, the health care provisioning model where the same company provides both public primary health care and OHS is not a typical setting in Finland overall but may appear in small cities. Even though the imbalance in sector representation and service provisioning was taken into account in every phase during the analysis and reporting of the results, it may diminish the general application of the results. The participant selection criteria of longer work relation and treating primarily adult patients was given to the chief physicians who were responsible for interviewee recruitment. However, chief physicians’ personal preferences may have influenced who was invited to participate.

These results could be used when executing the ongoing health care reform in Finland. We showed how personnel regard the cooperation between sectors, thus giving decision makers new perspective when health care provisioning is planned. When compared to the previous international research literature and the picture they draw, our results seem plausible. However, due to the research setting and the research aim, the generalization of these result is limited to small Finnish cities where the same health care provider carries out the public primary health care and the OHS.

The main researcher (L.V.) engaged in all research phases, from the preparation of the interview guide to reporting the results. The original citations were not modified or shortened, and citations were analyzed in Finnish until the reporting phase to avoid misinterpretations during the process. Throughout the study, the research group played an active role in the process commenting and guiding L.V., since he did not have previous experience with interviewing health care professionals. These actions among others were evaluated to increase the trustworthiness of the study, which is a more fitting term for qualitative research than reliability or validity [45].

The global COVID-19 pandemic also impacted this research process. Due to the pandemic, face-to-face interviews were switched to online meetings. This could have affected the contents and richness of description in the interviews, and created a feeling of haste in some interviews, especially in those where the camera had to be turned off. Furthermore, the acute pandemic crisis burdened interviewees, which may have affected their motivation to participate.

## 5. Conclusions

This study showed that even though cooperation between health care sectors was seen as beneficial, there appear to remain several obstacles to collaboration. Time was of the essence and the feeling of being rushed was described to affect cooperation and create difficulties in establishing cooperation. Additionally, poor knowledge about the other health care sector seemed to be linked to the formation of mutual trust. Mutual trust is also a major element in information exchange and without it there remains a risk that health care sectors will grow further apart. These aspects of cooperation in work ability issues should be acknowledged by the political decision-makers and executives of health care services when planning the provisioning of local health care services. This study showed that information transfer and mutual trust are key for multi-professional cooperation in patient work disability issues. Without mutual trust, the information transfer remains vague and does not enhance the patients’ care path. Thus, information transfer and availability should receive greater attention alongside increasing the health care professionals’ knowledge about the other sector and mutual trust. Practical solutions, such as developing the usability of current IT systems, would allow fluent and transparent information exchange between sectors. The current reform of health and social services in Finland provides an opportunity to prevent the siloing of sectors. To enhance the effectiveness of work disability prevention, national health policies should focus on the barriers and facilitators to the cooperation of different service sectors, including OHS.

## Figures and Tables

**Table 1 ijerph-19-11916-t001:** Interviewees’ demographics and interview information.

Interview	Interview Location	Interview Length (min)	City	Organization	Number of Interviewees	Profession	Gender
1	MS Teams	26	1	Public primary health care 1	1	GP	M
2	MS Teams	35	1	Public primary health care 1	1	GP	M
3	MS Teams	25	1	Public primary health care 1	1	GP	F
4	Meeting room	42	1	Public primary health care 1	3 *	Nurse	F
5	MS Teams	29	1	Public primary health care 1	1	GP	F
6	MS Teams	20	2	Public primary health care 2	1	GP	M
7	MS Teams	24	2	Public primary health care 2	1	GP	F
8	MS Teams	31	2	Public primary health care 2	1	GP	M
9	MS Teams	37	2	Public primary health care 2	4 *	Nurse	F
10	MS Teams	28	2	Public primary health care 2	1	GP	M
11	MS Teams	37	1	OHS 3	2 *	Occupational nurse	F
12	MS Teams	48	2	OHS 4	2 *	Occupational nurse	F
13	MS Teams	31	2	OHS 4	1	Occupational health physicians	M
14	MS Teams	26	3	Public primary health care 5	2 *	Nurse	F
15	MS Teams	36	3	Public primary health care 5	1	GP	F
16	MS Teams	21	3	Public primary health care 5	1	GP	M
17	MS Teams	30	3	Public primary health care 5	1	Chief physician	F
18	MS Teams	34	4	Public primary health care 6	1	Nurse	F
19	MS Teams	40	3	Public primary health care 5	1	GP	M
20	MS Teams	60	5	Public primary health care 7	2 *	Nurse	F

* All participants were of the same gender and profession.

**Table 2 ijerph-19-11916-t002:** An example of the analysis.

Data Extract	Code	Sub Theme	Theme	Topic
**Lack of connection** and phone call: I think we have a paper somewhere that has all the phone numbers… (searches the desk) I’m looking through the lists in my room to see whether there is a number to occupational health care anywhere... I don’t see it in the lists that are available to me here.	Communication	Information exchange conventions	Exchange of information	Cooperation between public primary health care and occupational health care on issues related to patient’s work ability
**Transition:** When you talk with one, the next you hear is that he or she is not working there anymore. Even our chief physician has changed.	Transition	Feeling of insufficiency	Resources for cooperation	Cooperation between public primary health care and occupational health care on issues related to patient’s work ability
**Length of reception:** But the situations can come as a surprise, a fifteen-minute appointment may have been scheduled for the patient, which is really a short time,—sometimes you may have to say that we need more time for example for tomorrow, so that we can really try to solve the situation as a whole.	Hurry	Feeling of insufficiency	Resources for cooperation	Cooperation between public primary health care and occupational health care on issues related to patient’s work ability

**Table 3 ijerph-19-11916-t003:** The coding tree.

**Topic**				
Cooperation between public primary health care and occupational health care on issues related to patient’s work ability				
**Themes**				
**Attitudes toward the other health care service sector**		**Exchange of information**		**Resources for** **cooperation**
**Sub Themes**		**Sub Themes**		**Sub Themes**
**Knowledge about the other health care** **service**	**Trust in the other stakeholder’s** **services**	**Information** **exchange** **conventions**	**Need for changes in cooperation**	**Feeling of** **insufficiency**
**Codes**		**Codes**		**Codes**
**Knowledge** (Unclear + Uncertainty + Unsure)	**Trust** (Service quality + Trust)	**Patient lost in the system** **Law** (Text availability + Missing information + Privacy policy) **Patient’s** **responsibility** **Communication** (Phone call + Direct contact + Direct connection + Making connection + Way of making connection + Information transfer + Information exchange + Communication + A good information exchange)	**Reduction of** **cooperation** (Decline of cooperation + Change of cooperation + Weakening cooperation) **Multiprofessional need** (Network meeting + Multiprofessional + Multiprofessional need + Multiprofessional requirement + Multiprofessional assessment + Multiple stakeholders) **Same service** **provider** (Same service provider + Defining responsibilities)	**Hurry** (Resources + Hurry) **Physical** **proximity** **Transition** (Change of employment + Reason (transition) + Transition)

## Data Availability

The data presented in this study are available on request from the corresponding author. The data will be publicly available in the Finnish Social Science Data Archive (https://www.fsd.tuni.fi/en/ accessed on 17 May 2022) as soon as all the manuscripts referring to the TYKYTUO ESF project are ready for publication (latest 1 January 2024).

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
