# Peer review of "Cooperation between Public Primary Health Care and Occupational Health Care Professionals in Work Ability-Related Health Issues"

_ijerph, 2022, doi:10.3390/ijerph191911916_

Round 1

Reviewer 1 Report (Previous Reviewer 4)

Thank You once again for having the opportunity of reviewing the paper. The authors did some significant work to address most of the issues I mentioned last time.

Although I'm generally satisfied with those extensions, I still encourage authors to try to make more systematic and in-depth comparisons with other similar studies in the relevant world literature.

I think that some extension of discussion and more in-depth comparison with the similar situation in other countries would increase the scientific value of the presented results.

I would also encourage the authors to elaborate more on the design of future studies that could be based on the outcomes of the present research.

I generally recommend the paper to be published in the International Journal of Environmental Research and Public Health journal given the above-mentioned issue is properly addressed.

Author Response

Dear Reviewer, please see the attachment.

We highly appreciate your comments and time.

Reviewer 2 Report (New Reviewer)

Dear Authors 

Thank you for providing such an interesting reading. 

Please provide information on why Occupational Therapists were not included in your study. 

Please clarify if the coding tree was developed by the authors, before or after the interviews, or if you have previous research on this topic that led you to these Topics/Themes and Sub-themes.

Author Response

Dear Reviewer, please see the attachment.

We highly appreciate your comments and time.

This manuscript is a resubmission of an earlier submission. The following is a list of the peer review reports and author responses from that submission.

Round 1

Reviewer 1 Report

This study is based on researcher selected interviews with 29 health professionals in five small Finnish cities that are unknown to readers. Clear theoretical starting points and research setting are missing. The questions posed to the interviewees are unknown. The applicability of empirical results has not been assessed and the validity and reliability of empirical research have not been evaluated. The shortcomings are fundamental and serious. This paper thus would have to be rejected.

Reviewer 2 Report

Thank you for submitting your paper. It is well-written, methods are properly described and results consistently drawn.

I would suggest to mention the Work Ability Index as it is a relevant tool in this field and also it could support information exchange and it is a useful resource for cooperation at the organisational level. Please consider mentioning it in both the introduction, discussion and/or as a strategy for policymakers in this field or for future research. (https://academic.oup.com/occmed/article/57/2/160/1584972)

Reviewer 3 Report

I would like to thank the authors for their work. 

This is an interesting paper, which aims to understand how professionals experience the cooperation between public primary health care and occupational health services regarding patients’ work ability.

The topic is updated. The introduction provides a strong rationale; the objectives of the research are clear. The methodology is robust and fit with the aims of this work. The results are consistent, and the conclusions are not speculative.

Reviewer 4 Report

Review of the Manuscript ID Number: ijerph-1818732 for the International Journal of Environmental Research and Public Health journal.

Title: Cooperation between public primary health care and occupational health care professionals in work ability-related health issues

I would like to thank the authors and editors for having had the opportunity to review this manuscript.

The research regards the problem of cooperation between public primary health care and occupational health care professionals. This qualitative study seems to be quite interesting, and it fits well into the scientific profile of the International Journal of Environmental Research and Public Health journal. The research is sufficiently grounded in the relevant literature, though some more references to investigations related to other countries could be beneficial to quality of the paper.

From the methodological point of view, the study is quite well prepared, however the interviews might cover more than only five small cities (<10,000 inhabitants) in Finland. The authors could at least discuss if the specific selection of participants would allow to generalize the obtained results to other small cities in Finland or other countries.

I think that some extension of discussion and more in-depth comparison with the similar situation in other countries would increase the scientific value of the presented results.

I would also encourage the authors to elaborate more on the design of future studies that could be based on the outcomes of the present research.

I generally recommend the paper to be published in the International Journal of Environmental Research and Public Health journal given the above-mentioned issues are properly addressed.